# ^1^H-NMR Metabolomics Analysis of the Effect of Rubusoside on Serum Metabolites of Golden Hamsters on a High-Fat Diet

**DOI:** 10.3390/molecules25061274

**Published:** 2020-03-11

**Authors:** Li Li, Manjing Jiang, Yaohua Li, Jian Su, Li Li, Xiaosheng Qu, Lanlan Fan

**Affiliations:** 1School of Pharmacy, Guangxi University of Chinese Medicine, Nanning 530001, China; cattmily@163.com (L.L.); jmj074321@163.com (M.J.); 000295@gxtcmu.edu.cn (Y.L.); 000682@gxtcmu.edu.cn (L.L.); 2National Engineering Laboratory of Southwest Endangered Medicinal Resources Development, Guangxi Botanical Garden of Medicinal Plants, Nanning 530023, China; 3Guangxi Scientific Research Center of Traditional Chinese Medicine, Guangxi University of Chinese Medicine, Nanning 530001, China; 000641@gxtcmu.edu.cn

**Keywords:** rubusoside, ^1^H-NMR, metabolomics, fat metabolism

## Abstract

Rubusoside is a natural sweetener and the active component of *Rubus suavissimus*. The preventive and therapeutic effect of rubusoside on high-fat diet-induced (HFD) serum metabolite changes in golden hamsters was analyzed by ^1^H-NMR metabolomics to explore the underlying mechanism of lipid metabolism regulation. ^1^H-NMR serum metabolomics analyses revealed a disturbed amino acid-, sugar-, fat-, and energy metabolism in HFD animals. Animals supplemented with rubusoside can partly reverse the metabolism disorders induced by high-fat diet and exerted good anti-hypertriglyceridemia effect by intervening in some major metabolic pathways, involving amino acid metabolism, synthesis of ketone bodies, as well as choline and 4-hydroxyphenylacetate metabolism. This study indicates that rubusoside can interfere with and normalize high-fat diet-induced metabolic changes in serum and could provide a theoretical basis to establish rubusoside as a potentially therapeutic tool able to revert or prevent lipid metabolism disorders.

## 1. Introduction

Lipid metabolism disorders affect the accumulation and distribution of lipids and their metabolites in the body. At present, the prevalence in China of lipid metabolism disorders (dyslipidemia)—an important risk factor for developing cardiovascular disease—among people above 18 years is 40.4% [1] while the number of cardiovascular patients exceeds 290 million. To replace chemically synthesized drugs, many research groups are increasingly looking for safe and effective active ingredients from natural or traditional Chinese medicines to treat lipid metabolism disorders.

Rubusoside, the main active ingredient of Guangxi sweet tea made from the leaves and shoots of *Rubus suavissimus* S. Lee, consists of a tetracyclic diterpene glycoside combined with alcohol and glucose (Appendix A) [2,3,4]. It is a low calorie natural sweetener with sweetness similar to white sugar [5] and about 300 times that of sucrose. In fact, the sweetness of 1 kg of rubusoside is equivalent to that of 15 kg of sucrose. Since rubusoside can replace saccharin and sucrose in food, medicine, and other industries, it is of special interest for patients with diabetes and obesity.

Being documented in the Guangxi Zhuang Medicine Standard (Guangxi Zhuang Autonomous Region Food and Drug) [6], Guangxi sweet tea has long been used by the Zhuang and Yao people of Guangxi for the treatment of heat and polydipsia, chronic pharyngitis, diabetes, and obesity. Consumption of rubusoside results in reduced fat [7,8,9,10] and sugar levels in the blood [11,12,13] as well as reduced blood pressure [12]. It has antibacterial [14] and anti-caries effects [15] and results in a reduced body weight due to limited caloric intake [16].

At present, research on rubusoside mainly focuses on whole animal physiology while its effect on in vivo metabolism is hardly understood. Modern instrumental analysis methods such as nuclear magnetic resonance spectroscopy and mass spectrometry allow qualitative and quantitative detection of endogenous and exogenous metabolites in biological fluids. As such, metabolomics analyzes life phenomena by establishing a more holistic view which could provide opportunities for cross-referencing in multiple directions in the medical field [17,18,19].

Considering that the lipid metabolism of male Syrian golden hamsters is quite similar to that of humans [20], we adopted this animal model in the current study. We applied ^1^H nuclear magnetic resonance (^1^H-NMR) metabolomics and quantitative analysis methods to analyze the effect of rubusoside on the serum of golden hamsters with high-fat and high-cholesterol diet-induced lipid metabolism disorder. A better understanding of the mode of action of rubusoside might ultimately provide a basis for its further development and broader utilization.

## 2. Results

### 2.1. Animal Experiment Results

According to literature, a dose of 250 mg/(kg·day) rubusoside significantly reduces serum triglyceride levels with 30.08% in rats on a high-fat diet [7], whereas a dose of 300 mg/(kg·day) rubusoside reduces blood glucose levels of streptozotocin-induced diabetic rats while stimulating insulin secretion [11]. However, as these doses are much higher than the normal daily dose in humans [6], we calculated the actual content of rubusoside in *Rubus suavissimus* S. Lee based on HPLC data [21] and adopted a dose for golden hamster of 60 mg/(kg·day).

To test the effect of rubusoside on high-fat diet-induced metabolic changes, we chose to apply two scenarios: (1) supplementation of rubusoside from the first day of the high-fat diet to test the so-called “preventive” effect of rubusoside on high-fat diet-induced lipid metabolism changes (HFD+PRbs); (2) supplementation of rubusoside from the third week of the high-fat diet to test the so-called “treatment” effect of rubusoside on high-fat diet-induced lipid metabolism changes (HFD+Rbs).

After ten weeks on a high-fat diet, animals in HFD showed a significant increased body weight, significantly higher levels of total cholesterol (TC) and triglycerides (TG) in the liver, and significantly higher levels of TG, TC, low-density lipoprotein cholesterol (LDL-C), high-density lipoprotein cholesterol (HDL-C), alanine transaminase (ALT), and aspartate transaminase (AST) in serum compared to animals in normal diet group (ND) (Figure 1), indicating that our high-fat diet model was well established [20,22]. Compared to HFD, animal in simvastatin (HFD+SV) group had significantly lower body weight and improved animal physiological and biochemical indicators of serum and liver high-fat diet except for LDL-C, HDL-C, and ALT (Figure 1).

Compared to HFD, animals in HFD+PRbs had significantly lower body weight, significantly lower levels of TC and TG in liver, and significantly higher levels of HDL-C as well as significantly lower levels of TC, TG, LDL-C, AST, and ALT in serum (Figure 1).

Immediate supplementation of rubusoside during high-fat diet was superior to delayed supplementation of rubusoside or simvastatin since, compared to HFD, animals in HFD+Rbs still evidenced similar liver TC and serum HDL-C (Figure 1B,G) whereas animals in HFD+SV evidenced similar serum LDL-C, HDL-C and ALT (Figure 1F–H), respectively. These data suggest that, especially with prophylactic administration (HFD+PRbs), rubusoside can outperform simvastatin in reducing fat deposition in the liver and in normalizing lipid metabolism-related indicators in the blood.

Compared to ND, HFD animals had larger livers with a yellowish brown color instead of dark red and a significantly greasier tissue cut surface. Except for a pink color, livers of HFD+SV, HFD+PRbs, and HFD+Rbs animals appeared similar to those of HFD (Appendix A). While there were no fat particles in liver sections of ND, liver section oil red O staining visualized a large number of orange-red fat particles in liver of HFD (Figure 2). Compared to HFD, liver sections of HFD+SV, HFD+PRbs, and HFD+Rbs had significantly fewer fat particles (Figure 2). Of note, the number of fat particles in HFD+PRbs was lower than that of the HFD+Rbs. Consistently, quantification of hepatic steatosis severity according to the integrated density revealed that the groups HFD+SV, HFD+PRbs, and HFD+Rbs had significantly less hepatic steatosis compared to ND (Figure 2F). These data indicate that, similar to simvastatin, both immediate (HFD+PRbs) as well as delayed (HFD+Rbs) supplementation of high-fat diet with rubusoside have anti-hepatic steatosis effects.

### 2.2. ^1^H-NMR Metabolomics Analysis

A total of 60 metabolites were detected in the serum of golden hamsters. The ^1^H-NMR spectra (Appendix A) were analyzed and classified in amino acids and derivatives, organic acids, nucleic acid components, sugars, and other components (See Table 1 for details).

#### 2.2.1. Analysis of Serum Metabolites

After normalizing the obtained data with the Pareto scaling method [23,24], we analyzed the data with unsupervised principal component analysis (PCA), a multivariate statistical method which reduces the number of observations through dimensionality reduction into a few comprehensive indicators while it preserves the main information of the original variables (Figure 3A). PC 1 and 2 explain 36.2% and 19.5% of the metabolic variance of serum, respectively. In the PCA score map, ND and HFD were well separated from each other indicating metabolic differences between both groups. Next, the data were clustered and visualized in a heatmap (Figure 3D). Similar to the PCA score map, ND and HFD were clustered separately indicating differences between both groups in the content of the detected metabolites.

Compared to HFD, HFD+PRbs and HFD+Rbs both situated in different locations in the PCA score map while partially overlapping with ND. This indicates that, while being on a high-fat diet, the metabolism of rubusoside-supplemented animals is more closely related to that of ND. Of note, the earlier the intervention with rubusoside during high-fat diet, the more similar the metabolite levels were compared to ND.

Next, we analyzed the data with partial least square discriminant analysis (PLS-DA), a statistical method which by emphasizing the differences between groups while minimizing the differences within the group better grasps the overall characteristics and variation of multidimensional data. The first two latent variables of the PLS-DA model cumulatively accounted for 45.5% of the total variance (first latent variable (LV1) with 28.4% and LV2 with 17.1%, respectively) and contributed to most of the variation. Cross validation showed a parameter R2 value of 0.865 and Q2 value of 0.746. A permutation test of 100 times revealed the observed statistic *p* < 0.01. Similar to the PCA score map, ND and HFD were well separated from each other, whereas HFD+PRbs and HFD+Rbs partially overlapped and were more closely related to ND (Figure 3B).

Based on PLS-DA, ANOVA, and fold change (FC), the variable importance in projection (VIP) scores revealed multiple significant, high-fat diet-induced alterations in HFD compared to ND (Figure 3C and Table 1). For example, compared to ND, HFD had higher levels of multiple amino acids including glutamic acid, glutamine, phenylalanine, serine, glycine, methionine, tryptophan, and leucine; higher levels of the tricarboxylic acid cycle (TCA) components 2-oxoglutarate, fumarate, and malate; higher levels of the ketone bodies acetone and 3-hydroxybutyrate; and higher levels of choline, 1,3-dihydroxyacetone, dimethylamine, lactate, and acetate. Compared to ND, HFD had significantly lower levels of formate, urea, and betaine, and lower levels of glucose. Combined, these data indicate that a high-fat diet disturbs the amino acid-, sugar-, energy-, and fat metabolism.

Although the levels of metabolites changed similarly in HFD+Rbs and HFD+PRbs compared to HFD, the preventive effect of rubusoside on high-fat diet induced metabolic changes was not as pronounced in HFD+Rbs compared to HFD+PRbs. For example, immediate supplementation with rubusoside at the onset of the high-fat diet (HFD+PRbs) resulted in significantly lower levels of the amino acids tyrosine, phenylalanine, glutamine, glutamic acid, alanine, serine, glycine, methionine, tryptophan, and leucine; significantly lower levels of the TCA components 2-oxoglutarate, fumarate and malate; significantly lower levels of the ketone bodies 3-hydroxybutyrate, choline, 1,3-dihydroxyacetone, and acetate; and significantly higher levels of urea and 4-hydroxyphenylacetate compared to HFD alone (Figure 3C and Table 1).

Delayed supplementation of rubusoside during the high-fat diet (HFD+Rbs) resulted in significantly lower levels of tyrosine, alanine, serine, tryptophan and leucine; significantly lower levels of the TCA components 2-oxoglutarate and malate; significantly lower levels of the ketone bodies choline, 1,3-dihydroxyacetone, acetate; and significantly higher levels of urea and 4-hydroxyphenylacetate (Table 1 and Figure 3C).

Of note, whereas 4-hydroxyphenylacetate levels were below detection limit in both ND and HFD, the average content in HFD+PRbs and HFD+Rbs was 0.0143 and 0.0030, respectively.

Taken together, these data indicate that rubusoside significantly interferes with and prevents multiple high-fat diet-induced metabolite changes.

#### 2.2.2. Analysis of Serum Metabolite Pathways

Metabolic pathway analysis revealed multiple alterations in HFD compared to ND. The altered metabolic pathways with the highest impact (impact > 0.1, *p* < 0.05) included amino acid-, energy-, sugar-, and lipid-related pathways including alanine, aspartate and glutamate metabolism; synthesis and degradation of ketone bodies; glycine, serine and threonine metabolism; pyruvate metabolism; arginine and proline metabolism; TCA cycle; beta-alanine metabolism; and aminoacyl-tRNA biosynthesis (Appendix A).

Considering the close overlap of HFD+PRbs with ND in PCA and PLS-DA, the overall characteristic metabolites of HFD+PRbs were closer to the metabolites in ND. We therefore decided to focus on HFD+PRbs and compare its metabolic pathway profile with HFD (Figure 3F and Table 2). The metabolic pathways with the highest impact (impact > 0.1, *p* < 0.05) included alanine, aspartate and glutamate metabolism; synthesis and degradation of ketone bodies; glycine, serine and threonine metabolism; arginine and proline metabolism; aminoacyl-tRNA biosynthesis; methane metabolism; butanoate metabolism; histidine metabolism; d-glutamine and d-glutamate metabolism; inositol phosphate metabolism; phenylalanine metabolism; tryptophan metabolism; tyrosine metabolism; lysine biosynthesis; and glycolysis or gluconeogenesis. These data suggest that supplementation of rubusoside from the onset of high-fat diet prevented many of the high-fat diet-induced changes in for example amino acid-, sugar-, and fat metabolic pathways. (Figure 3F and Figure 4).

## 3. Discussion

The current study analyzed the preventive effect of rubusoside on high-fat diet-induced alterations in lipid metabolism in golden hamsters. Animals on a high-fat diet had increased body weight, altered liver morphology as well as increased levels of TC and TG in the liver and enhanced levels of TC, TG, AST, ALT, HDL-C, and LDL-C in serum, indicating that our high-fat diet model was well established [20,22].

^1^H-NMR indicated high-fat diet-induced alterations in amino acid-, energy-, sugar-, and lipid-related metabolism. For example, a high-fat diet disturbed amino acid metabolism ultimately altering the levels of glutamate, glutamine, phenylalanine, serine, glycine, methionine, and tryptophan. Since amino acids are important in multiple metabolic pathways functioning as enzyme substrates or as biochemical regulators, changes in their levels might affect the metabolic and functional status of the body. For example, the glycine-serine-threonine metabolic pathway converts serine into pyruvate, an important energy metabolism precursor of the TCA cycle. Changes in tryptophan levels affect the nervous system [25]. As glutamate and glutamine are precursors of the antioxidants glutathione and taurine, they represent important amino acids in the process of metabolic diseases [26]. Parallel with those amino acid metabolism disturbances, the urea content decreased further suggesting an abnormal liver function [27,28].

In addition, animals in HFD had increased levels of 2-oxoglutarate, fumarate, and malate compared to ND, indicating a deregulated TCA cycle. Next to being the core of cellular energy metabolism, the TCA cycle is also an important chemical synthesis route for glucose, amino acids, and fatty acids. Thus, conforming with previous literature data [29], increased fat intake resulted in depleted blood glucose levels. Under conditions of glucose deficiency, extrahepatic tissues use ketone body oxidation to sustain their energy demands which could explain the observed increased acetone and 3-hydroxybutyrate ketone body formation.

Under aerobic conditions, pyruvate is converted to acetyl-CoA by decarboxylation which then enters the TCA cycle. However, due to a hyperlipidemia-induced decrease in dissolved oxygen in the serum, acetyl-CoA decarboxylation is slowed down resulting in increased pyruvate levels. As both pyruvate and 1,3-dihydroxyacetone are intermediates in glucose metabolism [30], the observed elevated levels of 1,3-dihydroxyacetone and pyruvate in HFD compared to ND suggest a deficient glucose metabolism.

Furthermore, HFD showed increased choline levels compared to ND. Since choline indirectly participates in the fat metabolism and transportation in the body, promotes the utilization of fatty acids in the liver, accelerates the export of TG from the liver, and prevents liver damage [31], increased choline levels are indicators of disturbed fat metabolism.

When supplemented during a high-fat diet, rubusoside not only prevented high-fat diet-induced gain in body weight but also preserved liver morphology as well as the levels of many of the lipid metabolism markers in liver and serum as shown in Section 2.1.

^1^H-NMR metabolomics suggested that supplementation of rubusoside from the onset of high-fat diet prevented many of the high-fat diet-induced changes in, for example, amino acid-, sugar-, and fat metabolic pathways, (Figure 3F and Figure 4) indicating that rubusoside exerts a beneficial effect on high-fat diet-induced metabolic alterations (Figure 4).

4-hydroxyphenylacetate is a marker for preservation or modulation of the gut microbiota [32]. HFD was previously shown to alter gut microbiota [33] and to decrease 4-hydrophenylacetate levels [26]. Thus, considering the enhanced levels of 4-hydroxyphenylacetate in the rubusoside-supplemented groups, it is tempting to speculate that rubusoside might, at least partially, exert its protective effect on high-fat diet-induced metabolic changes through an effect on gut microbiota as was previously shown for Fushuan brick tea [34]. Along this line, the fact that HFD+PRbs had almost five times higher levels of 4-hydroxyphenylacetate compared to HFD+Rbs (0.0143 vs. 0.0030) could be an underlying reason prophylactic administration outperforms delayed supplementation of rubusoside during HFD. However, this should be addressed in future studies.

## 4. Materials and Methods

### 4.1. Materials

Rubusoside and simvastatin were purchased from Chengdu Pufei De Biotech Co. Ltd. (Chengdu, China) and Hangzhou MSD Pharmaceutical Co., Ltd. (Hangzhou, China), respectively. Male LVG hamsters (110–130 g, eight weeks old) were purchased from Vital River Laboratory Animal Technology Co., Ltd., Beijing, China and used under animal license number SCXK (Beijing) 2012-0001. High-fat diet consisted of 2% cholesterol + 24.5% lard + 73.5% basic feed and was purchased from Beijing Keao Xieli Feed Co., Ltd.

### 4.2. Animal Experiment

Animal care and procedures were approved by and conducted according to the standards of the Guangxi University of Chinese Medicine (Nanning, China). Animals were maintained in a humidified (50–70%), temperature-controlled (22–25 °C) room on a 12 h:12 h light-dark cycle with food and water ad libitum.

To test the effect of rubusoside on high-fat diet-induced metabolic changes, we chose to apply two scenarios: (1) supplementation of rubusoside from the first day of the high-fat diet to test the so-called “preventive” effect of rubusoside on high-fat diet-induced lipid metabolism changes (HFD+PRbs); (2) supplementation of rubusoside from the third week of the high-fat diet to test the so-called “treatment” effect of rubusoside on high-fat diet-induced lipid metabolism changes (HFD+Rbs). Thus, after one week of adaptive feeding, the animals were randomly divided into five groups with five animals in each group: normal diet group (ND), high-fat diet group (HFD), “prevention of rubusoside” group (60 mg/kg·day, HFD+PRbs), “treatment of rubusoside” group (60 mg/kg·d, HFD+Rbs), and the simvastatin group (2.5 mg/kg·day, HFD+SV). The ND group was fed with normal feed. In the HFD+PRbs group, rubusoside was intragastrically administered from the onset of the high-fat diet. In the HFD+Rbs group and the HFD+SV group, rubusoside or simvastatin were intragastrically administered from the third week after the onset of high-fat diet. Animals in HFD were given the same dose of carboxy methyl cellulose sodium solution. Body weight was measured weekly for a total of ten weeks.

At the end of the experiment, the hamsters were anesthetized with 3% pentobarbital sodium (0.3 mL/100g) after fasting for 12 h and blood samples were taken from the heart. The blood was left standing for 30 min, centrifuged at 3000 rpm for 10 min and the supernatant collected. Levels of TC, TG, LDL, and HDL were measured with the Automatic Analyzer (HITACHI 7600, Hitachi High-Tech Co., Ltd., Tokyo, Japan). The liver of each group was weighed, frozen sections were taken, and oil red O staining was used to mark the fat. Then Image J software was used to measure the integrated density of the red particles [35]. The remaining samples were frozen in liquid nitrogen and stored at −80 °C. The dosage of rubusoside was calculated according to the daily dosage of *R. suavissimus* used by the normal human body and then converted according to its content in *R. suavissimus* (5%) [6].

### 4.3. Serum Sample Processing

One milliliter of serum sample was centrifuged at 13,000 rpm for 10 min and the resulting supernatant was centrifuged at 13,000 rpm for 40 min at 4 °C for ultrafiltration membrane centrifugation. Next, 450 µl of the filtrate was added to 50 µl of 3-(trimethylsilyl) propane sulfonic acid (4.078 mM) for 10 s and centrifuged at 13,000 rpm for 2 min at 4 °C. Finally, 480 µL of the supernatant was taken in a nuclear magnetic tube.

### 4.4. ^1^H-NMR Spectroscopy

Serum NMR experiments were performed on a Bruker AV III 600 MHz spectrometer (Bruker Bio spin, Corporation, Billerica, MA, USA) equipped with an inverse cryoprobe operating at 600.13 MHz as our described method [22]. All NMR spectra of samples were acquired using a standard Bruker noesygppr1d pulse sequence and a total of 64 scans were collected into 32,768 data points over a spectral width of 8000.00 Hz. The first increment of a 2D-^1^H, ^1^H-NOESY pulse sequence was utilized for the acquisition of ^1^H-NMR data and for suppressing the solvent signal. Experiments used a 100 ms mixing time along with a 990 ms pre-saturation (~80 Hz gammaB1), and recycle delay is 1s. Spectra were collected at 25 °C, with a total of 128 scans over a period of 15 min. The data was analyzed using Chenomx software (Version 8.1, Alberta, AB, Canada) to conduct phasing and baseline correction with a consistent setting. The signal at δ 4.76–4.97 (water) in serum samples was excluded. Spectral data in the region of δ 0.50 to δ 10.00 were then normalized using the total spectral area normalization method and binned into a bin spectral width of 0.04 ppm. Spectral intensities were then scaled to sodium 3-(trimethylsilyl) propionate-2,2,3,3-*d*4 (TSP). Metabolic profiling and peak identification were performed by Anachro Technologies Inc. (Wuhan, China) [22,36,37].

### 4.5. Statistical Analysis

The data analysis methods were described previously [22]. The results of animal physiological and biochemical indicators are presented as mean ± standard deviation (SD). T-test was used to compare the two groups. The multi-group comparisons were analyzed by one-way analysis of variance (ANOVA) with SPSS 24.0 (SPSS Inc., Chicago, IL, USA). Values of *p* < 0.05 were considered significant. Multivariate data analysis of log-transformed and Pareto scaling and the serum metabolomics data was performed using MetaboAnalyst 4.0 (Xia Lab at McGill University, Montreal, QC, Canada). The distribution and relative quantification of the identified metabolites were analyzed by Ward’s Hierarchical Clustering and visualized with a heatmap. Variable importance in projection (VIP) produced by PLS-DA, ANOVA, and fold change (FC) were applied to discover the contributable variable for classification. Finally, the variables with VIP > 1, *p* < 0.05 and FC ≥ 2 were treated as important metabolites. For the metabolic pathway analysis, the original data including all the metabolites of samples was used. The pathway analysis was performed with MetaboAnalyst (www.metaboanalyst.ca/) [38,39].

## 5. Conclusions

In conclusion, ^1^H-NMR serum metabolomics provide a powerful approach to explore the possible mechanism underlying the lipid metabolism regulation of rubusoside. As revealed by the golden hamster model, rubusoside prevented the disturbances induced by high-fat diet in amino acid-, sugar-, energy-, and fat metabolism to varying degrees, and intervened in amino acid metabolism, synthesis of ketone bodies, as well as choline and 4-hydroxyphenylacetate metabolism. Importantly, compared with delayed supplementation, supplementation of rubusoside from the onset of high-fat diet showed better effects, suggesting a preventive effect of rubusoside on high-fat diet-induced lipid metabolism disorder in golden hamsters. As such, considering that the lipid metabolism of male Syrian golden hamsters is quite similar to that of humans, our data might provide a theoretical basis to establish rubusoside, a naturel sweetener, as a therapeutic remedy able to revert or, with a prophylactic daily dose, prevent lipid metabolism disorders. As such, it may have great prospects in human daily food and medical treatment.

## Figures and Tables

**Figure 1 molecules-25-01274-f001:**
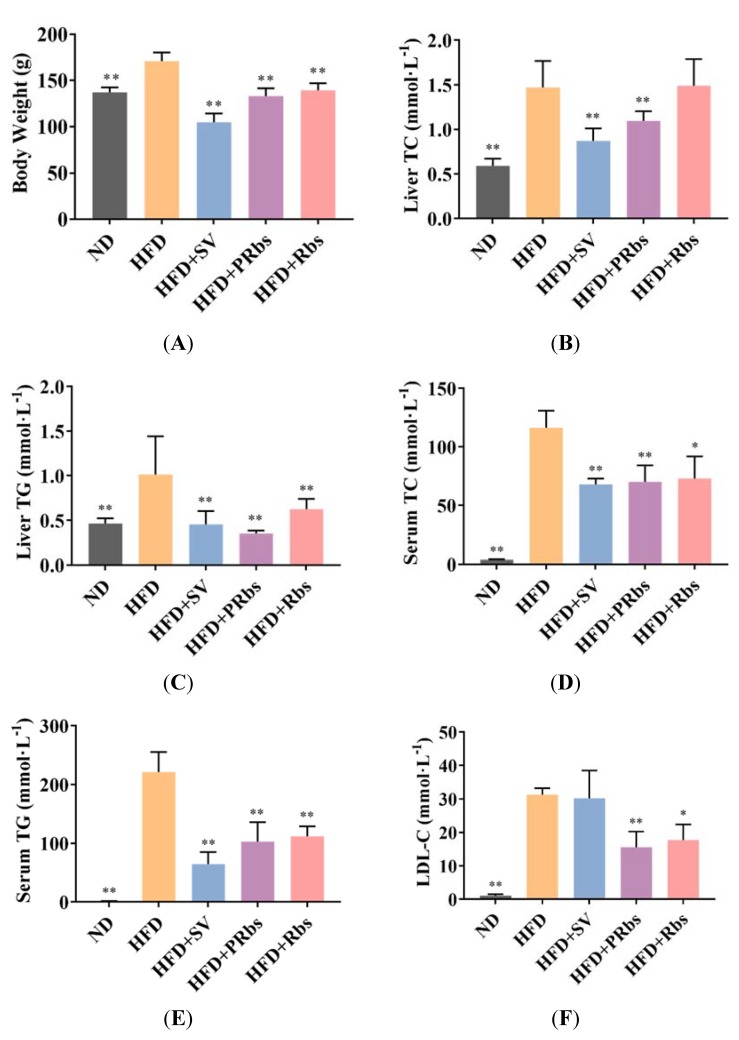
Effect of high-fat diet and high-fat diet supplemented with simvastatin (HFD+SV), preventive rubusoside (HFD+PRbs), or treatment rubusoside (HFD+Rbs) on body weight (**A**), liver TC (**B**) and TG (**C**), serum TC (**D**), TG (**E**), LDL-C (**F**), HDL-C (**G**), ALT (**H**), and AST (**I**). Data are presented as mean ± SD * *p* < 0.05, ** *p* < 0.01 compared to HFD; *n* = 5. TC, total cholesterol; TG, triglycerides; LDL-C, low-density lipoprotein cholesterol; HDL-C, high-density lipoprotein cholesterol; ALT, alanine transaminase; AST, aspartate transaminase.

**Figure 2 molecules-25-01274-f002:**
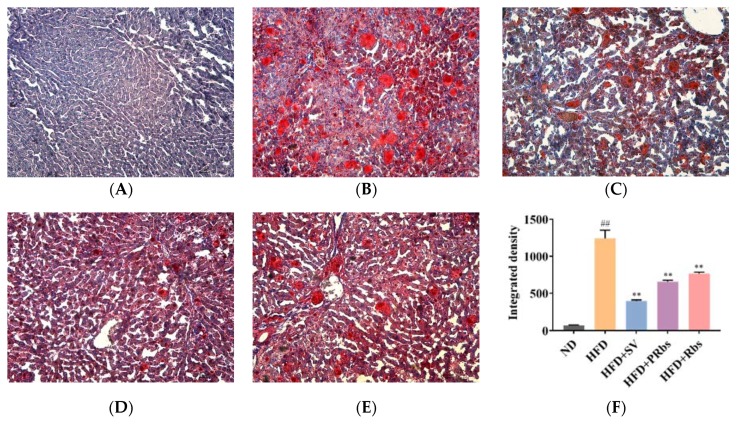
Effects of rubusoside on hepatic lipid deposition illustrated by histological analysis: oil red O staining was applied to liver sections from different experimental animal groups: (**A**) ND; (**B**) HFD; (**C**) HFD+SV; (**D**) HFD+PRbs; (**E**) HFD+Rbs; (**F**) quantitative analysis of Oil red O staining of liver sections. Oil red O staining results in visualization of neutral fat in red. ^##^ compared with ND, *p* < 0.01; ** compared with HFD, *p* < 0.01.

**Figure 3 molecules-25-01274-f003:**
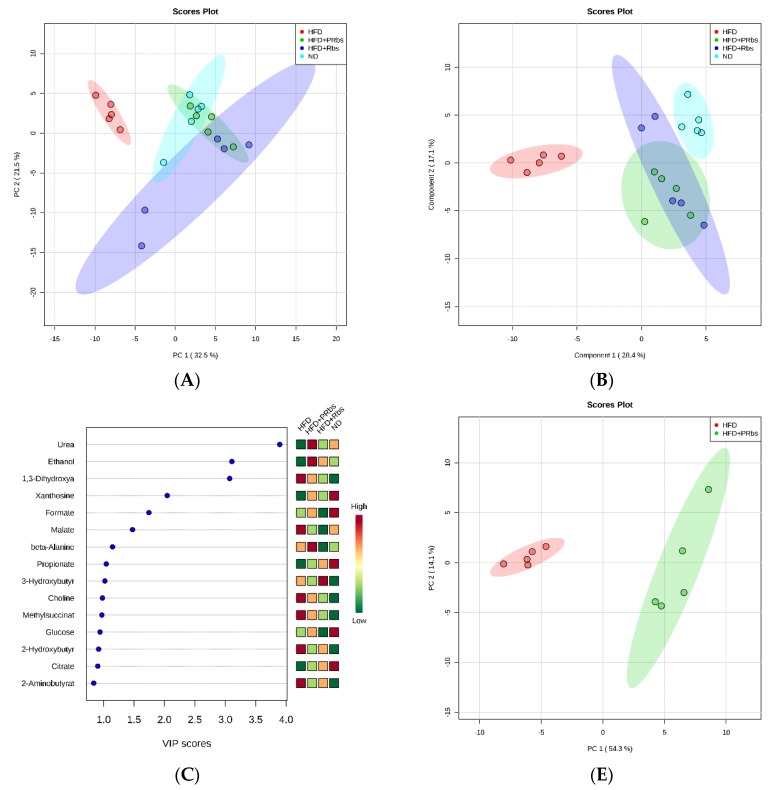
^1^H-NMR metabolism analysis of hamster serum. (**A**) PCA score, (**B**) PLS-DA score, (**C**) PLS VIP, and (**D**) clustering analysis and heatmap visualization of ND vs. HFD vs. HFD+PRbs vs. HFD+Rbs; (**E**) PCA score, and (**F**) pathway analysis of HFD vs. HFD+PRbs.

**Figure 4 molecules-25-01274-f004:**
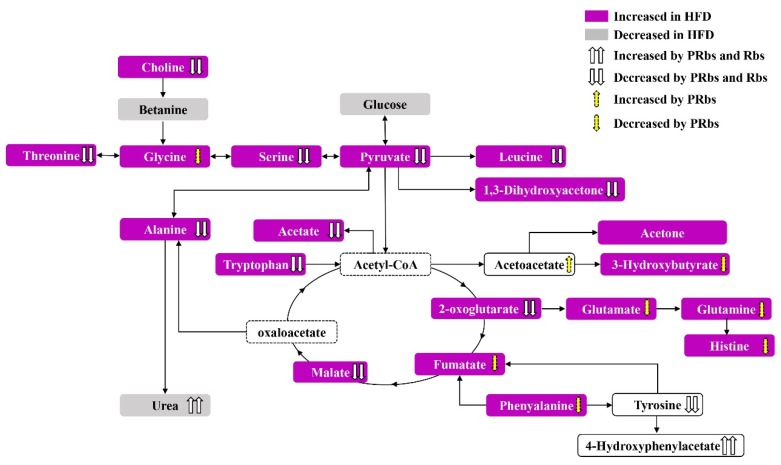
Summary of the metabolomic pathway analysis in the golden hamster serum. See text for details.

**Table 1 molecules-25-01274-t001:** Trends of metabolites concentration in serum of hamsters.

Label	HFD/ND	HFD+PRbs/HFD	HFD+Rbs/HFD
1,3-Dihydroxyacetone	32.06 **	0.06 **	0.03 **
2-Aminobutyrate	2.09 *	0.57 **	0.63
2-Hydroxybutyrate	4.26 **	0.60 *	0.80
2-Oxoglutarate	3.68 **	0.39 **	0.38 *
2-Oxoisocaproate	2.39 **	0.76 *	0.91
3-Hydroxybutyrate	4.69 **	0.73 *	1.09
3-Hydroxyisobutyrate	0.93	0.71 *	1.03
4-Hydroxyphenylacetate	≈0 ^a^	>0 ^a^	>0 ^a^
Acetate	1.53 *	0.50 **	0.56 *
Acetoacetate	0.91	4.21 *	4.94
Acetone	2.21 **	1.44	1.46
Alanine	1.34 *	0.59 **	0.48 **
Arginine	1.13	0.74	0.53
Aspartate	1.42	1.09	1.31
Betaine	0.50 **	1.75	1.45
Carnitine	1.68	0.80	0.47 *
Choline	4.10 **	0.45 **	0.34 **
Citrate	0.48 **	0.92	0.99
Creatine	1.97 **	1.18	1.55
Dimethyl sulfone	0.54 **	1.73 **	1.23
Dimethylamine	2.10 **	1.10	2.49
Ethanol	0.04	27.51 **	35.81
Ethylmalonate	1.79 *	0.67 **	0.95
Formate	0.16 **	0.89	0.81
Fumarate	1.46	0.58 **	0.48
Glucose	0.46 **	0.98	0.75
Glutamate	1.86 *	0.68 *	0.83
Glutamine	1.39 *	0.69 **	0.71
Glycine	2.29 **	0.71 **	0.63
Histidine	1.84 *	0.69 *	0.79
Isoleucine	1.39	0.72 *	0.79
Lactate	1.29 *	0.74	0.74 *
Leucine	2.17*	0.53 *	0.63 *
Lysine	1.00	0.76	0.89
Malate	2.35 **	0.42 **	0.23 **
Mannose	2.15 **	0.77	0.61
Methanol	1.38	0.86	1.23
Methionine	1.18	0.79 *	0.82
Methylsuccinate	2.91 **	0.67 *	0.68
Nicotinurate	3.09	0.13 *	0.00 *
O-Phosphocholine	1.88	0.73	0.67
Phenylalanine	1.90 **	0.75 *	0.80
Proline	1.00	0.46 **	0.81
Propionate	0.57	1.11	2.56
Pyroglutamate	3.42 **	0.43 **	0.59 *
Pyruvate	1.95 *	0.40 **	0.29 **
Serine	1.27 *	0.63 **	0.39 *
Succinate	1.26	1.25	0.68
Threonine	1.40	0.48 **	0.45 *
Trimethylamine	1.57	1.20	1.89
Tryptophan	1.50 **	0.57 **	0.56 **
Tyrosine	1.10	0.62 **	0.48 **
Uracil	10.44	0.59	0.24
Urea	0.17 *	13.37 *	14.49 **
Uridine	1.65 *	0.71	0.23 **
Valine	1.18	0.78 **	0.91
Xanthosine	0.07 **	7.91	2.34
myo-Inositol	1.30 *	1.54	2.05
sn-Glycero-3-phosphocholine	1.68 *	0.92	0.72
beta-Alanine	1.87	1.33	0.45

* compared with HFD, *p* < 0.05; ** compared with HFD, *p* < 0.01; ^a^ the detected levels of 4-hydroxyphenylacetate were below detection limit in both ND and HFD, but above detection limit in HFD+PRbs and HFD+Rbs. The numbers indicate the ratio of the average of the indicated groups.

**Table 2 molecules-25-01274-t002:** Metabolic pathway analysis of the serum of golden hamster (HFD vs. HFD+PRbs).

Pathway Name	Total Compounds	Hits	Raw *p*	-Log (*p*)	Holm Adjust	FDR	Impact
Alanine, aspartate and glutamate metabolism	24	8	0.01	4.63	0.25	0.02	0.71
Synthesis and degradation of ketone bodies	6	3	0.00	7.48	0.03	0.00	0.70
Glycine, serine and threonine metabolism	48	9	0.00	10.29	0.00	0.00	0.44
Arginine and proline metabolism	77	9	0.00	7.91	0.02	0.00	0.33
Aminoacyl-tRNA biosynthesis	75	18	0.00	9.53	0.00	0.00	0.23
Methane metabolism	34	7	0.00	8.84	0.01	0.00	0.20
Butanoate metabolism	40	7	0.00	6.81	0.05	0.00	0.17
Histidine metabolism	44	4	0.00	5.82	0.11	0.01	0.14
d-Glutamine and d-glutamate metabolism	11	3	0.01	5.29	0.16	0.01	0.14
Inositol phosphate metabolism	39	1	0.00	5.69	0.12	0.01	0.14
Phenylalanine metabolism	45	6	0.04	3.21	0.84	0.06	0.12
Tryptophan metabolism	79	1	0.00	7.08	0.04	0.00	0.11
Tyrosine metabolism	76	6	0.01	5.27	0.16	0.01	0.11
Lysine biosynthesis	32	3	0.00	5.84	0.11	0.01	0.10
Glycolysis or Gluconeogenesis	31	5	0.00	6.61	0.06	0.00	0.10

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
