# Peer review of "^1^H-NMR Metabolomics Analysis of the Effect of Rubusoside on Serum Metabolites of Golden Hamsters on a High-Fat Diet"

_molecules, 2020, doi:10.3390/molecules25061274_

Round 1

Reviewer 1 Report

The study conducted by Li and coworkers entitled "1H NMR metabolomics analysis of the effect of rubusoside on serum metabolites of golden hamsters on a high-fat diet” is an interesting and well performed work.

Minor points”

The authors presente in the Figure 2. Figure 2. Oil red O staining of liver sections indicated that rubusoside improved hepatic lipid deposition, however, this statement is based only in the image? Is it possible to add quantitative analysis and statistics to better describe this results?

The authors used T-test for group comparison. Why did the group choose this test? Since the study have several groups, wouldn’t be more appropriate to use ANOVA and post tests?

The authors performed metabolic pathways analysis including only metabolites in statistical significant difference? Wouldn’t more informative if two analysis were performed, one for the unregulated and a second for the down regulated?

Reviewer 2 Report

The study applied 1H NMR-based spectroscopy to analyze the metabolic effects of a natural supplement, Rubus suavissimus, on a high-fat diet hamster. Indeed, the results presented here provide insights into the regulation of lipid metabolism.

Overall, the manuscript is well-written with a concise description of the results and discussions; but weak in the experimental methods. Minor revision should be considered before publish in Molecules.

 Here are some comments:

-what labelled Figure ‘S1’? I suppose this meant to be in the supplementary material. In fact, the supplemental materials are missing entirely in this manuscript.

-since NMR is the primary tool for the study, it should offer some representative spectra of each group, even as supplementary materials. This gives confidence in the quality of the spectral data.

-Figure 1: SV (I suppose simvastatin group). There is a lack of discussions (in the text) on SV, as if there is no reason to include the data.

-Figure 2: I suppose these are the ‘histology’ with oil red O staining?

-lines 95-97: if possible, provide photo-images on these descriptions.

-Figure 3: there are no quality parameter(s) provided for the PCA (component, R2, Q2, etc…) and PLS (cross-validation, etc…).

-line 136: add ‘, ‘ before whereas

-line 223-225: should backup this statement

Material & Methods: (1) no much information (size, dimension, gender, age etc…) was provided on the animal before the treatment; (2) NMR: what are the acquisition parameters: mixing time and Recycle delay? And also the parameters of the spectral processing; (3) nearly zero description on your data analysis: matrix-size (chemical range and bucket-size); spectral normalization and scaling before the analysis?

Conclusions: (1) the word ‘superior’ is probably overstated. Base on figure 1, it does not seem the effects with PRbs are much better than those with Rbs. (2) expand the conclusions with a brief description on the significance ‘findings’ on the metabolites and on the lipid metabolism. (3) could these results link (i.e. translational info) to human dietary?
